# Automating data analysis for hydrogen/deuterium exchange mass spectrometry using data-independent acquisition methodology

Frantisek Filandr[1,8], Vladimir Sarpe[1,8], Shaunak Raval[1,2,8], D. Alex Crowder[1], Morgan F. Khan[1], Pauline Douglas[1], Stephen Coales[3], Rosa Viner [4], Aleem Syed [5], John A. Tainer[6,7], Susan P. Lees-Miller [1] & David C. Schriemer [1,2] ✉

We present a hydrogen/deuterium exchange workflow coupled to tandem mass spectrometry (HX-MS[2]) that supports the acquisition of peptide fragment ions alongside their peptide precursors. The approach enables true auto-curation of HX data by mining a rich set of deuterated fragments, generated by collisional-induced dissociation (CID), to simultaneously confirm the peptide ID and authenticate MS[1]-based deuteration calculations. The high redundancy provided by the fragments supports a confidence assessment of deuterium calculations using a combinatorial strategy. The approach requires data-independent acquisition (DIA) methods that are available on most MS platforms, making the switch to HX-MS[2] straightforward. Importantly, we find that HX-DIA enables a proteomics-grade approach and wide-spread applications. Considerable time is saved through auto-curation and complex samples can now be characterized and at higher throughput. We illustrate these advantages in a drug binding analysis of the ultra-large protein kinase DNA-PKcs, isolated directly from mammalian cells.

Hydrogen/deuterium exchange coupled to mass spectrometry (HX-MS) is a labeling method based on the exchange between protein backbone amide hydrogens and deuterated labeling buffers. HX-MS is used to obtain information about higher-order protein structure and dynamics, as the rate of amide hydrogen exchange is influenced by local structure and solvent accessibility. For example, it can help investigate protein folding mechanisms[1], discover ligand binding sites, and highlight allosteric effects of binding[2]. In the biotherapeutics industry it is particularly useful for epitope mapping[3]. The use of native solution conditions during labeling and the low sample requirements of the method makes HX-MS an appealing technology for structure-function analysis. Even complex multi-protein systems can be interrogated, if the system produces a suitable number of peptides in the sample workup process. The method has been reviewed extensively in recent years[4–12]. The standardization of experimental and data reporting protocols have improved the

[1]Department of Biochemistry and Molecular Biology, University of Calgary, Calgary, AB T2N 4N1, Canada. [2]Department of Chemistry, University of Calgary, Calgary, AB T2N 4N1, Canada. [3]Trajan Scientific & Medical - Raleigh, Morrisville, NC, USA. [4]Thermo Fisher Scientific, San Jose, CA, USA. [5]Division of Radiation and Genome Instability, Department of Radiation Oncology, Dana-Farber Cancer Institute, Harvard Medical School, Boston, MA 02215, USA. [6]Department of Molecular and Cellular Oncology, The University of Texas MD Anderson Cancer Center, Houston, TX 77030, USA. [7]Molecular Biophysics and Integrated Bioimaging, Lawrence Berkeley National Laboratory, Berkeley, CA 94720, USA. [8]These authors contributed equally: Frantisek Filandr, Vladimir Sarpe, Shaunak Raval. ✉e-mail: dschriem@ucalgary.ca

accessibility of the technology, creating the reliable biophysical technique that we know today[13].

However, while HX-MS enjoys frequent use in protein analysis, it is generally restricted from applications involving high-throughput characterizations, or analyses involving protein states much larger than 150 kDa of unique sequence[6]. This restriction is not primarily due to limitations in the analytical systems. Advancements in ion mobility can reduce spectral complexity[14,15] and new methods even support nanoHX on ultra-large complexes[16]. Robotic technologies remove much of the complexity of data collection[17–19] and even sub-zero chromatography can extend the window of elution time and minimize the problem of deuterium back exchange[20].

The remaining major bottleneck is data analysis. Complex isotopic profiles are generated by the technique and even with the improvements mentioned above, spectral overlap still occurs and deuteration values can be miscalculated. These worsen as sample complexity increases. We seek good-quality mass spectra with clean peptide isotopic envelopes when collecting deuteration data. To date, the field manually inspects the raw data to curate spectral selections and discard peptide signals that are compromised in any way. Retention time is used to confirm the identity of a peptide, supported by the accurate mass of a deuterium-shifted signal and signal "quality" is visually assessed, relying upon years of experience in spectral assessment. But as the size of the protein system grows, or the number of states to screen increases, manual curation becomes impractical and such an approach has always been prone to human error. A solution is needed that would remove the burden of manual data curation entirely, while also tolerating more convoluted spectra arising from complex mixtures, even whole cell lysates. Unfortunately, all major data analysis engines only focus on supporting manual review and post-curation activities such as data visualization and statistical analysis[21–27].

One possible strategy for automated analysis involves leveraging peptide fragmentation and the MS² domain. The acquisition of fragments in an HX-MS² experiment can corroborate the identity of a peptide (validation) and generate abundant data to confirm the deuteration level of the precursor peptide (authentication). The approach relies upon deuterium scrambling, which is ubiquitous under normal ion transmission conditions[28]. We had previously demonstrated the potential of such an approach[29]. However, we lacked an efficient way to collect and mine the data in a comprehensive and platform-independent manner.

In this work, we demonstrate how data-independent acquisition (DIA) can be used for HX-MS² experiments as a method to obtain deuteration data from both MS¹ and MS² domains simultaneously (Fig. 1). We adapt a RANSAC computer vision algorithm that allows us to automatically select the best isotopologues and even rescue overlapped signals. The resulting analyses are comparable to expert curated datasets, while offering objectively curated data and a clear measure of reliability for each peptide datapoint. Export-ready figures in the form of uptake plots and differential Woods plots are automatically generated at the end of data processing. We demonstrate that highly complex samples drawn directly from cell lysates can be accurately interrogated in minutes, rather than days or weeks.

## Results and discussion

### The properties of CID-generated HX-MS² data

The fragmentation of deuterated peptides is accompanied by gas-phase scrambling of deuterium in a manner that is dependent upon the energetics of ion transmission and the mode of fragmentation used[30,31]. Scrambling is extensive when using ion optics settings consistent with high transmission efficiency. Such settings impart considerable ion activation during desolvation and ion focusing[32]. Conventional CID fragmentation contributes additional thermal energy and ensures that scrambling is complete. Scrambling currently can only be overcome with a combination of milder ion transmission conditions together with electron-mediated fragmentation modes[28,32–34].

Scrambling engages all sources of labile hydrogens in a peptide, and a full atom accounting reveals a linear relationship between the deuteration of a fragment and the number of its labile hydrogen sites (Fig. 2). The fragment deuteration model for the given deuterium-labeled peptide highlights a typical fit. This linear model intersects the origin and, in the absence of any spectral overlap in the MS¹ domain, passes through the deuteration value of the precursor peptide. Selecting a single fragment is therefore sufficient to replace the precursor as an accurate and precise measure of deuteration, when scaled for size[29]. The longer and more intense fragment ions tend to be more sensitive measures of deuteration than smaller and less intense ones, but essentially all sequence ions can be used as surrogates for deuteration measurements, individually or combined. Only fragments that undergo neutral loss appear to deviate from the linear model, likely due to a kinetic isotope effect[29].

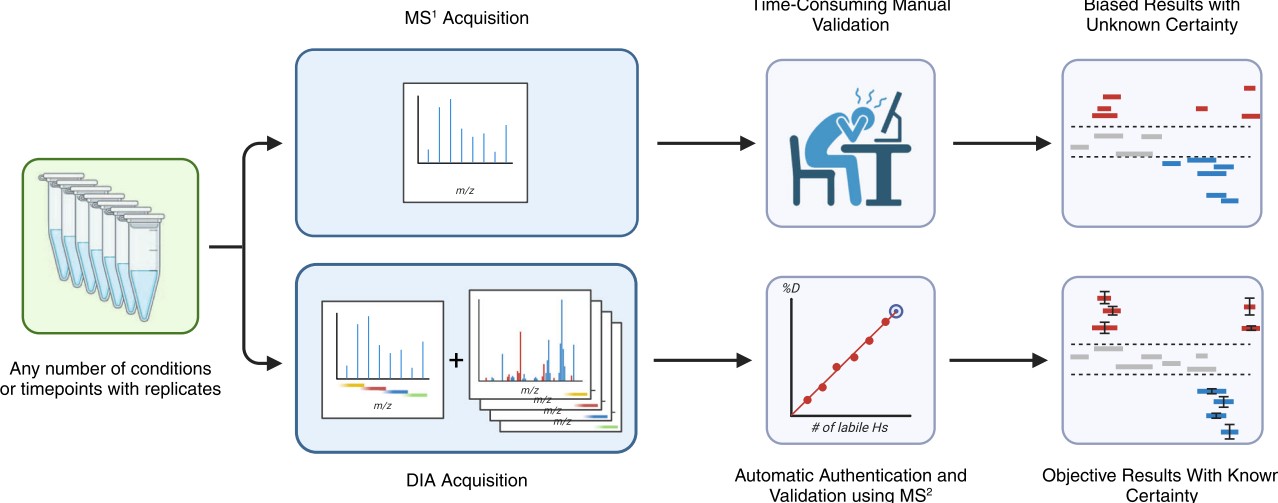

**Fig. 1 | DIA acquisition enables automation.** DIA data acquisition uses deuterium-scrambled CID or HCD fragments as surrogates that confirm the identity and the deuteration value of any given peptide (bottom workflow), replacing the traditional procedure that mines only MS1 data (top workflow). DIA supports the implementation of an automation approach to generate Woods plots or deuterium uptake curves without any user input required, while also offering data reliability for each peptide based on fragment statistics. Created with Biorender.com.

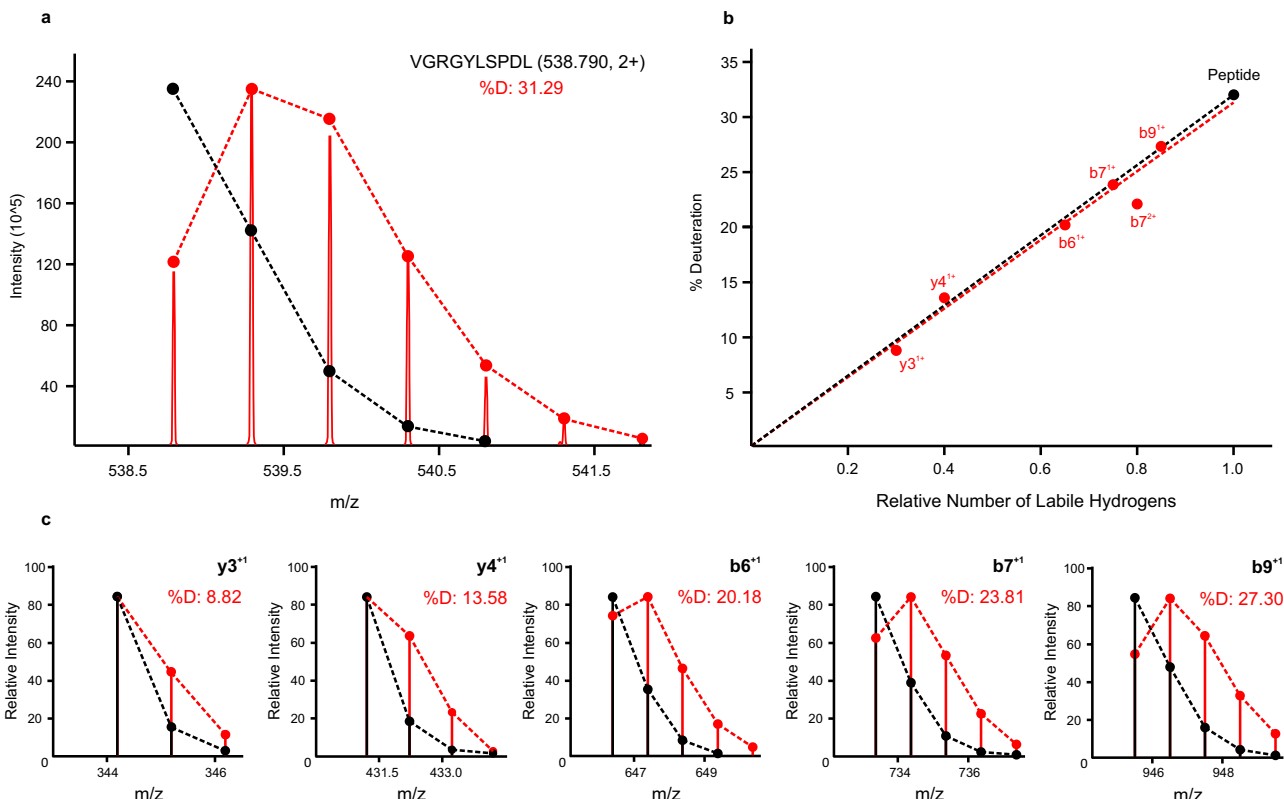

**Fig. 2 | Effect of hydrogen/deuterium scrambling on fragment deuteration values for a given peptide. a** MS¹ spectrum of deuterated VGRGYLSPDL (2+ ion), showing native (black) and deuterium-expanded (red) isotopic envelopes. **b** Corresponding fragment deuteration model, where red dots represent fragment ion deuteration values and the black dot represents the precursor peptide deuteration. **c** Select fragment isotopic distributions supporting the model, showing native (black) and deuterium-expanded (red) isotopic envelopes.

## Building a DIA-based HX-MS² workflow

Previous illustrations of fragment deuteration surrogacy used targeted MS² acquisitions and data-dependent acquisition (DDA) experiments, together with limited deuteration experiments (e.g., 20% D₂O labeling)[29]. Reduced deuteration ensured that ions could be sampled within a conventional small ion transmission window (e.g., m/z 2). A data-independent acquisition (DIA) experiment, with its wider transmission windows, should allow acquisition of fragment data for routine deuterium labeling experiments as we have previously suggested[35]. We refined our original HX-MS² concept to implement this strategy.

We applied a standard DIA method design with one exception. To minimize spectral complexity in the MS² spectra yet promote good sampling of chromatographic features, we restricted the mass range slightly and designed DIA ion transmission windows to be as small as possible for a given platform. The faster-scanning the instrument, the smaller the windows can be made. However, unlike standard DIA methods, we use larger overlaps between successive windows to ensure that strongly deuterated peptides have at least one window where the peptide isotopic distribution is not truncated by the edge of a transmission window (Fig. 3), as such a truncation would result in distortion of resulting fragment isotopic envelopes due to missing isotopologues, and cause errors in the deuteration readout. The degree of overlap used in the DIA experiment is selected based on the percentage of D₂O used in the experiment. A window overlap of m/z 4 was found to be sufficient for 50% D₂O labeling and did not significantly compromise cycle times on either the TOF or Orbitrap platforms used in this study. We note that more common labeling levels of 75-90% D₂O would require slightly larger overlaps (e.g. m/z 5-6) but we note parenthetically that such high percentages are not a requirement of the basic HX-MS technique[36].

We developed AutoHX, a software app in the Mass Spec Studio, to mine HX data in two dimensions. The software automatically selects the ideal, non-truncating DIA window for a given peptide and calculates deuteration values for the precursor peptide from the MS¹ data and for all fragments from the MS² data. The app currently requires a peptide library. This library is obtained from DDA runs that we collect at the beginning of an HX experiment, using matched but undeuterated control digests. These DDA runs can be searched with any standard peptide identification search engine. We revised HX-PIPE, our HX-tailored search engine[37], to generate the library. HX-PIPE finds unambiguous peptide assignments and then formats a library for direct use in AutoHX. It has the option of generating a specific set of transitions from the peptide library or deferring transition selection to AutoHX. We have found the latter to be more practical when fragmentation conditions vary slightly between the DDA and DIA runs. Extracted ion chromatograms are then generated for each peptide in the library, and a window of integration is defined to produce averaged MS¹ and MS² spectra for deuteration analysis. A set of filters is applied to parse low-quality signals from the dataset and then a RANSAC-based spectral analyzer is applied that selects the best set of isotopologues for all peptides and their fragments (Supplementary Fig. 1). This spectral analyzer selects peaks based on a chosen deuteration model. EX2 is the current default but a more complex EX1 model is also enabled. EX1 kinetics would also be carried over to the fragments in MS² space, and in extreme cases (e.g., a large peptide with high deuteration and labeling at 90% D₂O) the DIA window overlap may need to be widened slightly.

To determine the increase in peptide redundancy we obtain from a DIA-based deuteration measurement, we collected a triplicate, 6-timepoint kinetics analysis of phosphorylase B (a 97 kDa protein) in HX-DIA mode (Fig. 4). Analysis of the MS¹ space generated 380

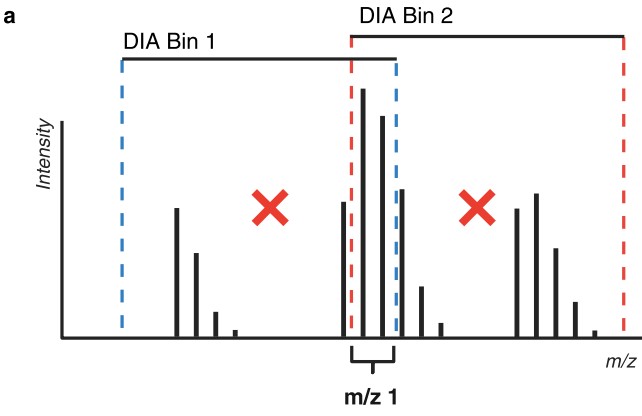

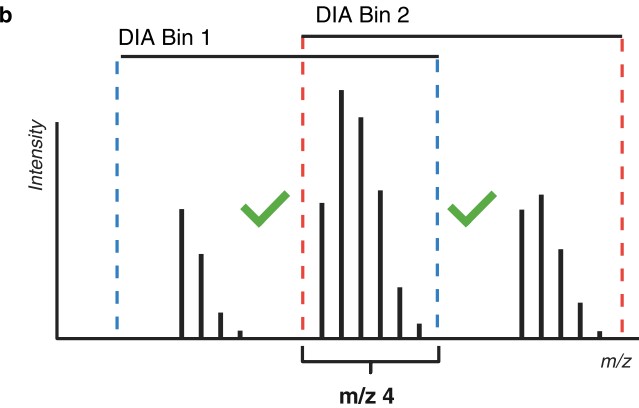

**Fig. 3 | DIA window overlap overview. a** The standard m/z 1 DIA window overlap used in proteomics is insufficient if the goal is to obtain full and intact isotopic envelopes of fragment ions. If any of the isotopologues of the precursor peptide ion are cut out of the DIA bin, the resulting fragment isotopic envelopes will be distorted. **b** An overlap of m/z 4 was found to be effective for a typical HX-MS experiment with most 2+ and 3+ charged peptides.

consistently useable peptide signals across all timepoints and replicates, whereas the fragment space generated 3269 usable fragment signals. Only peptides with 3 or more detectable quality fragments were used in this calculation. Thus, even though these data were collected on an older model instrument (QExactive Plus), adding the fragment dimension increases the redundancy in sequence coverage dramatically from 4.6 (MS[1] only) to 49.7 (MS[1] plus MS[2]). This extra redundancy does not improve resolution, but it does translate into higher-quality measurements. For example, in the Phosphorylase B dataset, including the MS[2] data decreases the standard deviation of the measured peptide deuteration by 41% compared to using just MS[1] data.

### Automated data authentication – kinetics

We noticed that there are instances where the MS[1] measurement generates a more precise peptide deuteration measurement, and instances where a single fragment or even a combination of fragments generates a better measure. To automatically generate kinetic curves using the best of the underlying data in terms of accuracy and precision, we developed a method where all possible combinations of MS[1] and MS[2] data for a given peptide are created, producing a normal distribution of deuteration values (Fig. 5). The distribution is sampled and the combinations closest to the mean are mined for the one that generates the most precise deuteration value across the replicates. This combination was chosen to represent the peptide and is used in developing the kinetics curve. The final combination may differ

between timepoints because the optimization step is done for every timepoint, to ensure that the cleanest signal is obtained.

Mining high-redundancy data in this manner conveys two benefits. First, the fragment data validates the peptide because we require a minimum number of unique fragments. Second, the distribution tests the authenticity of the deuterium calculation. A peptide that produces a narrow distribution indicates an accurate and precise measure of deuteration, whereas a peptide that generates a non-normal and/or wide distribution highlights a compromised measurement. The flawed feature is then discarded from the dataset. Peptide deuteration kinetics from the phosphorylase B experiment were generated using this automation strategy and compared to a carefully curated manual analysis of the MS[1] data (Fig. 6). The resulting heatmaps are almost indistinguishable, confirming that DIA-generated fragments can be used very effectively for automated peptide curation. The corresponding kinetics curves for all 380 peptides are provided in supplementary data (Supplementary Fig. 2). During development, we detected a very slight bias against peptides with comparatively poor fragmentation, such as short singly charged peptides. To rescue high-quality peptides in this category, we adopted a strategy from clinical mass spectrometry[38]. Qualifier transitions were required to validate peptide identity but were not used for deuteration calculations. Rather, they endorsed MS[1]-based deuteration measurements provided the latter were of high quality (i.e., high isotopic fidelity).

### Automated data authentication – differential analysis

HX-MS is most often used in a relational (or differential) manner. That is, deuteration kinetics for a protein in one state are compared to the same protein in a second state, either with a single labeling timepoint or an integration of the kinetic series. Common applications include drug or ligand characterization studies and quality control in the manufacture of protein biologics. These differential analyses are often depicted in a Woods plot, which shows induced changes in labeling as a function of protein sequence. To automate the generation of these plots, we developed a variation of the optimization method described above (Fig. 5). Here, the distribution is formed using the same combination strategy but the ΔD value is used instead. That is, given a specific combination of MS[1] and MS[2] data, average deuteration values are calculated from each replicate of a given state and compared to the average value from each replicate of the control state (Fig. 7). The distribution provides a solution to the problem of assigning significance to a given change in deuteration. The width of the distribution is used to assign a confidence interval to the change. An estimation of error overcomes the subjectivity of assigning relevance to a change based solely on its magnitude. This strategy provides a per-peptide measure of significance that has been lacking in the field, based on the subjectivity of manual data analysis. For example, a small change in a strongly ordered region of structure that takes up little deuterium can be classified as significant if the distribution is narrow. Peptides with conflicting deuteration values that cover areas of common sequence can be interpreted more rationally using the confidence interval as a guide.

### Differential analysis of Pol Θ drug binding

To test this automation approach, we applied it to a manually curated dataset collected in DIA mode but analyzed in the conventional MS[1]-only fashion[39]. Pol Θ, a DNA polymerase, is a cancer drug target. It is upregulated in 70% of breast and epithelial ovarian cancers and it contributes to mechanisms of resistance to both conventional and emerging therapies[40]. The antibiotic novobiocin was recently shown to bind to the protein[41]. We generated a sequence map of Pol Θ in the usual fashion and then conducted replicate HX-DIA analysis of novobiocin-bound vs free Pol Θ. The deuterated data were analyzed in three ways. First, we naïvely applied the full sequence map and

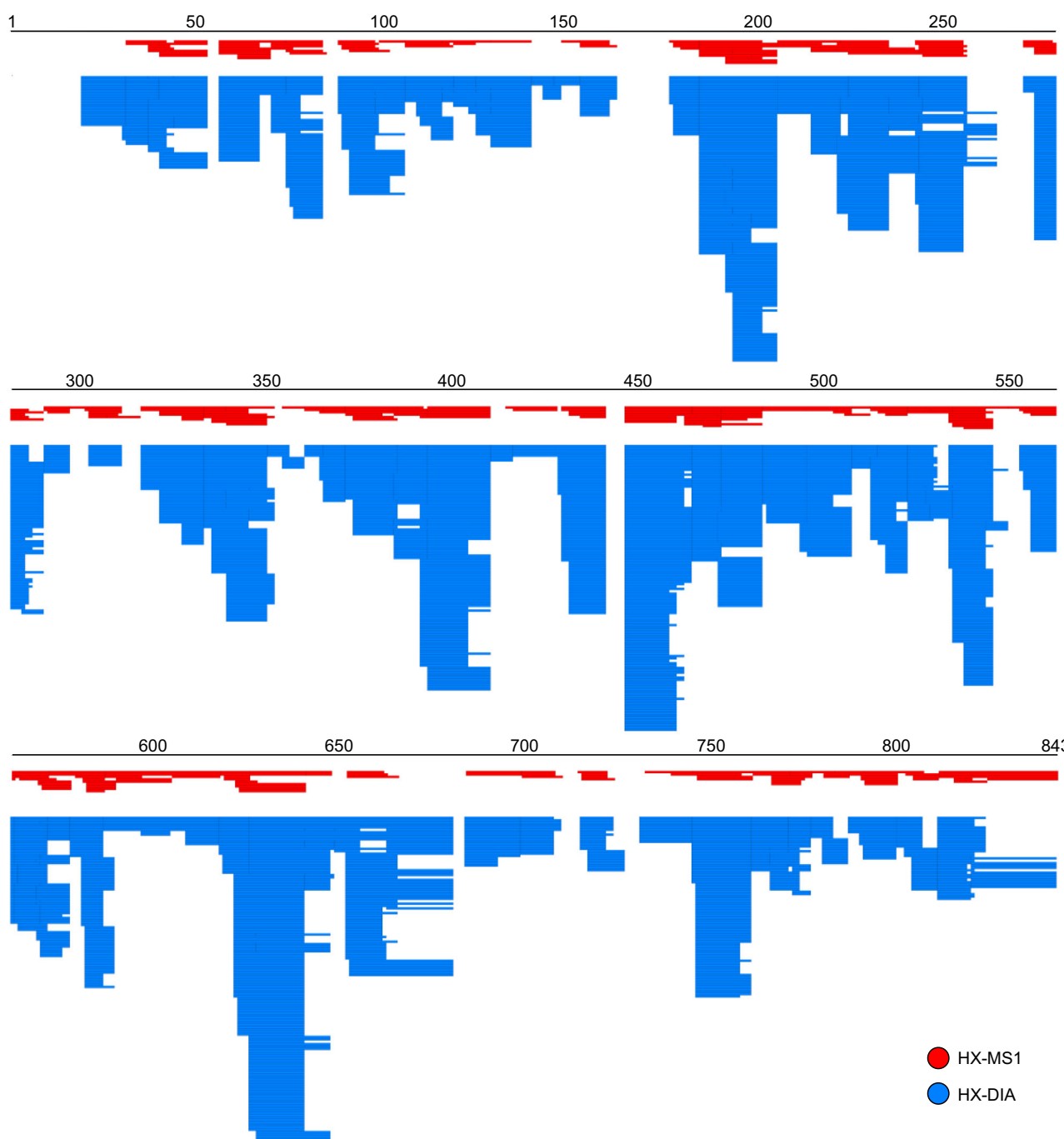

**Fig. 4 | DIA data expand deuteration maps.** Deuteration map resulting from an HX-DIA kinetics analysis of phosphorylase B, showing the large increase in the redundancy of coverage obtained by including each high-quality fragment in the analysis. Red represents the map from MS$^1$ data alone, and blue represents the map using both MS$^1$ and MS$^2$ data.

generated deuteration differences from the MS$^1$ domain data (Supplementary Fig. 3). Second, the MS$^1$ data were manually inspected by two experts and conflict-free peptides with good deuteration fit values were accepted (Fig. 8a). Third, AutoHX selected peptides and fragments automatically and determined the best subset to report. Woods plots were generated for each approach (Fig. 8b).

Not surprisingly, this exercise shows that curation is a key requirement in HX-MS analysis. Peptide IDs from a proteomics search do not guarantee good-quality peptides in HX-MS analysis. Manual curation is required in all existing HX-MS software packages to remove outliers and other suspicious values caused by retention time

misassignment and/or spectral overlap. Auto-curation using DIA data produces a map that is nearly identical to one generated from rigorous manual curation and indeed, it revealed some hidden biases in manual review. (For example, we detected changes in sequence regions already showing change, but were less diligent in their assessment. With the benefit of MS$^2$ data, we could see that some peptides were misidentified as they were not supported by fragment data.) No manual input was needed apart from assigning initial values to processing parameters tied to data quality, such as ppm errors in MS$^1$ and MS$^2$ and retention time precision. We note that this specific Pol Θ dataset was collected on a TOF instrument, which highlights

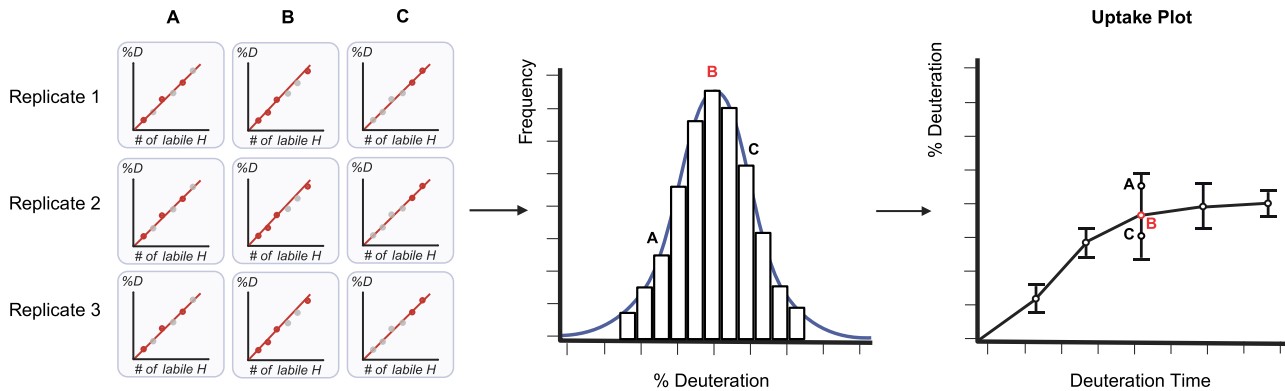

**Fig. 5 | Data combination concept for uptake curves.** After the selection of valid fragments and peptide distributions, shared across replicates and states, deuteration values calculated from all possible data combinations (illustrated as A, B, and C) are used to generate a deuteration value distribution. The mean of the distribution is selected as the reported deuteration value, and the width of the distribution (two standard deviations, user selectable) is used as an error bar on the uptake plot.

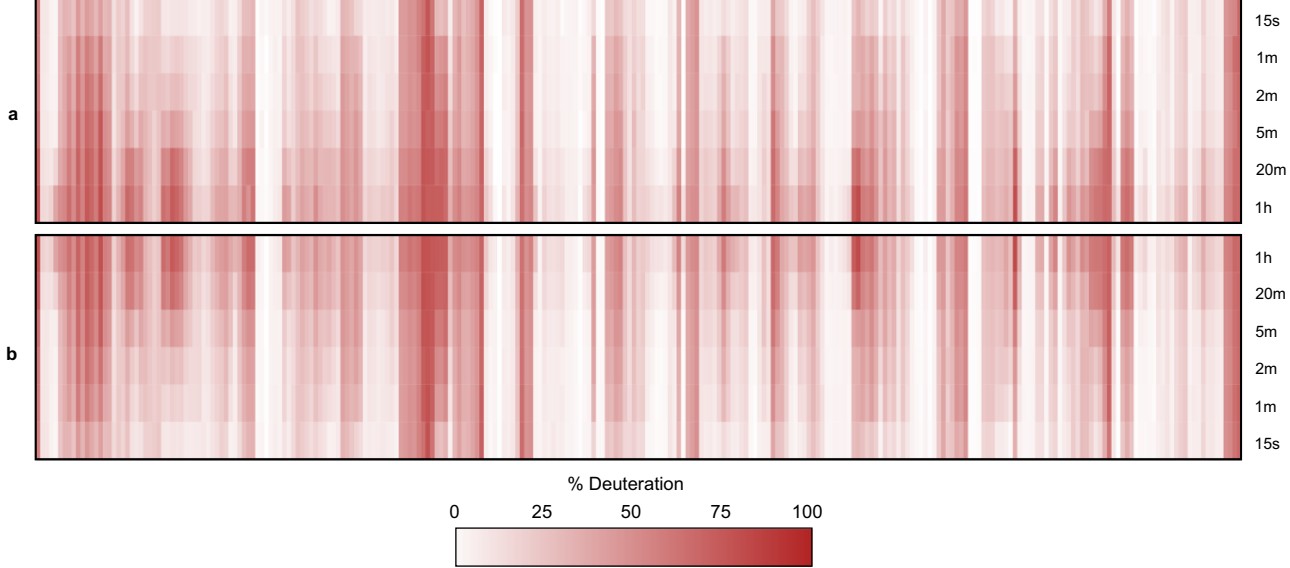

**Fig. 6 | Comparison of manual MS1 and AutoHX-derived deuteration values.** Data represents a 6-timepoint deuteration kinetics analysis of Phosphorylase B with all 380 peptide uptake plots shown as a heatmap ordered by sequence position. **a** AutoHX-derived deuteration values as described in the text. **b** MS1-derived deuteration values. A minimum of three useful fragments per peptide shared across all replicates and timepoints was set as a requirement for a peptide to be accepted.

the platform-independent nature of auto-curation routine. While not necessary or even encouraged, manual curation options are still retained in AutoHX.

**Differential analysis of drug binding to DNA-PKcs**

Automating data analysis creates opportunities for applications that previously were highly impractical. For example, using affinity isolates as input for HX-MS is very appealing, as it would avoid recombinant protein production and difficulties in reconstituting functional states. Affinity pulldowns are typically low yielding and even with extensive washing, target proteins are often co-isolated with a significant fraction of nonspecific binding proteins.

To test the performance of DIA and AutoHX on such a challenging sample type, we analyzed DNA-PKcs in a microscale pulldown experiment and used the isolate as input to a drug binding analysis. DNA-PKcs is a protein kinase that regulates double-strand DNA break repair. It is also one of the largest mammalian proteins (~450 kDa). It functions as a conformational switch at the point of commitment to the non-homologous end-joining repair pathway. Several experimental anticancer therapeutics target the ATP binding site, and only recently have their binding modes been modeled by cryo-EM[42]. The challenge, in part, involved isolation from over 100 L of cell culture equivalent to generate sufficient protein for analysis, given the difficulties associated with heterologous expression[42,43]. Here, we isolated GFP-tagged DNA-PKcs from the lysate from only two 10 cm plates of CHO cells, sufficient to generate enough material for the sequence map and six HX-MS experiments: three replicates of a drug-bound kinase and three ligand-free controls. We used AZD7648 as the drug, a selective inhibitor of DNA-PKcs that sensitizes cancer cells to radiation, doxorubicin, and olaparib[44].

We followed a typical HX-MS workflow with one exception. The higher complexity of the sample required a compositional analysis to build a searchable database and avoid false positive identifications. Using label-free quantitation, we detected 30 proteins that represent at least 95% of the sample. DNA-PKcs itself contributed 30–35% of the total signal and produced 2414 peptides. We then naïvely generated a Woods plot from the six HX-MS experiments using the entire peptide list (Fig. 9a), revealing a complexity that would take days or weeks of manual curation to correct. AutoHX was able to process all six samples in 10 min on a single high-end desktop computer.

The total number of usable peptides was reduced to 488, each with 4 or more unique fragments. This filter generated a sequence

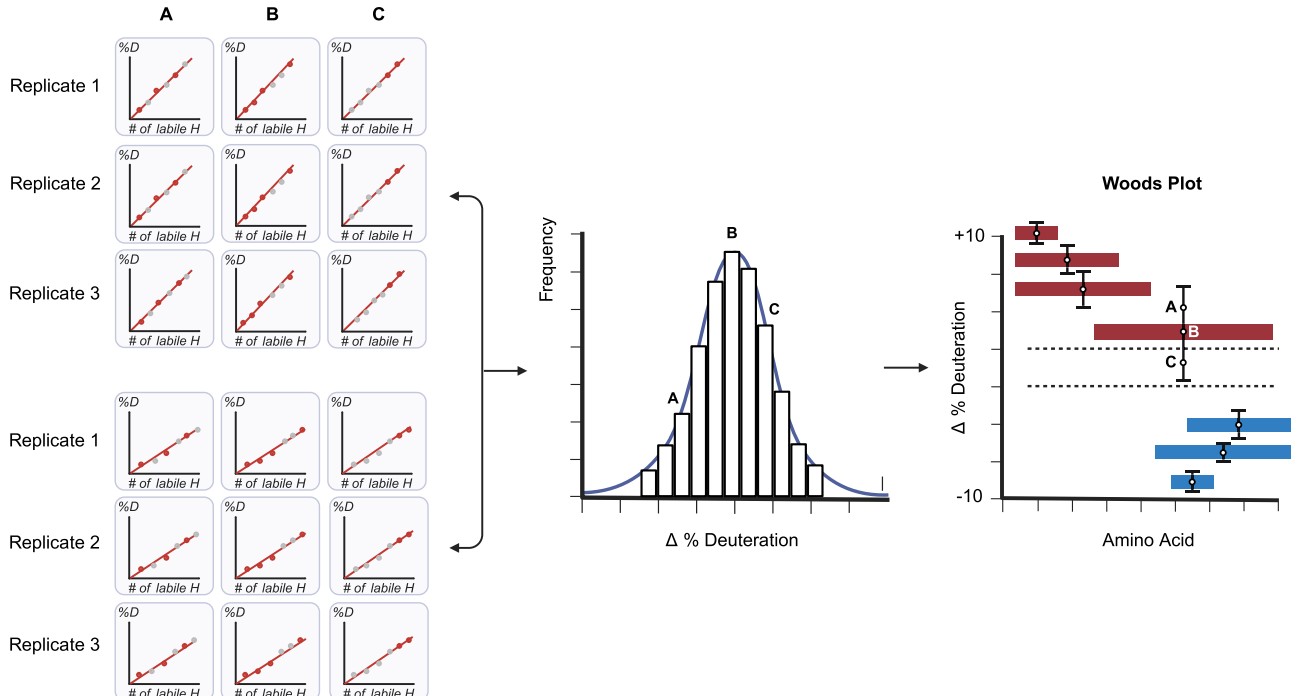

**Fig. 7 | Data combination applied to differential analysis.** Deuteration difference values between two states calculated from all possible data combinations (illustrated as A, B, and C) are used to generate a deuteration difference value distribution. This establishes a sampling precision for any measured change and facilitates data interpretation. Red represents induced destabilizations and blue induced stabilizations. The mean of the distribution is selected as the reported deuteration value, and the error bars represent two standard deviations (user selectable).

coverage of ~55% with strong coverage of the kinase domain, and widespread drug-induced stabilization of the protein is evident (Fig. 9b). This broad stabilization is anticipated. The control state is expected to be at least partially nucleotide-free and a previous HX analysis of the three-protein complex containing DNA-PKcs showed a similar effect arising from nucleotide binding[45]. The most confident changes were mapped onto the recent structure of AZD7648:DNA-PKcs, using only those peptides with error estimates outside of the noise limits (Fig. 9c). Interestingly, most of the detectable stabilizations are found in the FAT and kinase domains. One of the densest clusters identifies the hinge loop, which defines the primary binding site of the ligand[42]. Further optimization of the isolation should enhance this HX-MS assay and support the expansion of screening activities, and tailored DIA methods (e.g., variable window sizes and stepped collision energies) should enhance sequence coverage. We note that the conformational response of DNA-PKcs is critical to repair pathway commitment and is potentially druggable through allosteric inhibition. Accelerated HX-MS workflows should prove useful in exploring this concept.

In summary, automation tools have improved the rate at which HX-MS data can be collected, but the burden of manual data analysis has limited the extent to which the technology can be applied to many interesting problems. Deuterium scrambling offers a solution to the challenge of automating data analysis, by turning a problem into an opportunity. A manual inspection of the fragment models for all the peptides in the study did not show any evidence for retaining regioselective labeling. Scrambling indeed seems to be the norm. By invoking the DIA methodology for comprehensive fragment detection, AutoHX removes the burden of data analysis while simultaneously providing both peptide validation and data authentication. There is a parallel to be drawn with the development history of proteomics. Early methods for protein detection relied on extensive fractionation (e.g., 2D gels) followed by fingerprint-based MS-only methods using MALDI

TOF. The transition to MS/MS enabled the direct analysis of far more complex states, supported by complexity-tolerant search engines. HX-DIA provides a conceptually similar paradigm shift. It supports a proteomics-grade approach that should democratize a technology platform that has long been viewed as the domain of specialists.

## Methods
### HX-MS² of phosphorylase B
**D$_2$O labeling.** Phosphorylase B (Sigma-Aldrich, P6635) was resuspended in 40 μl HEPES buffer (25 mM, pH 7.4, 150 mM NaCl) to a concentration of 10 μM and diluted with 40 μl of deuterated HEPES buffer (25 mM, pD 7.4, 150 mM NaCl) to create a 50% D$_2$O labeling mixture. Labeling was conducted over multiple timepoints from 15 s to 1 h, and 12 μl aliquots were quenched 1:1 (v/v) with 250 mM glycine buffer (pH 2.3) containing 0.5 μg/μl nepenthesin II, resulting in a 1:1 protein:protease ratio (w/w). Samples were digested at 8 °C for 2.5 min and then flash frozen in liquid nitrogen. Samples were prepared in triplicate for each timepoint and thawed immediately prior to HX-MS² analysis of 20 μl injections. Samples for sequence mapping based on data-dependent acquisition (DDA) were processed in the same way, except D$_2$O was replaced with H$_2$O in the labeling phase.

**Data collection.** Data were acquired on a Thermo Scientific Q Exactive™ Plus mass spectrometer (with the Xcalibur 4.6 operating system) connected to a LEAP PAL HDX autosampler and a Thermo Scientific UltiMate™ 3000 LC system. All samples were prepared manually and injected into the cold compartment of the autosampler set to 4 °C. The injected peptides were trapped on a Luna 5 μm C18(2) 100 Å Micro Trap (20 × 0.50 mm, Phenomenex) and separated on a Luna Omega 3 μm Polar C18 100 Å LC Column (50 × 0.3 mm, Phenomenex) using a standard 5% to 45% solvent B gradient (10 min. gradient). Solvent A was 0.4% FA in H$_2$O and solvent B was 0.4% FA in

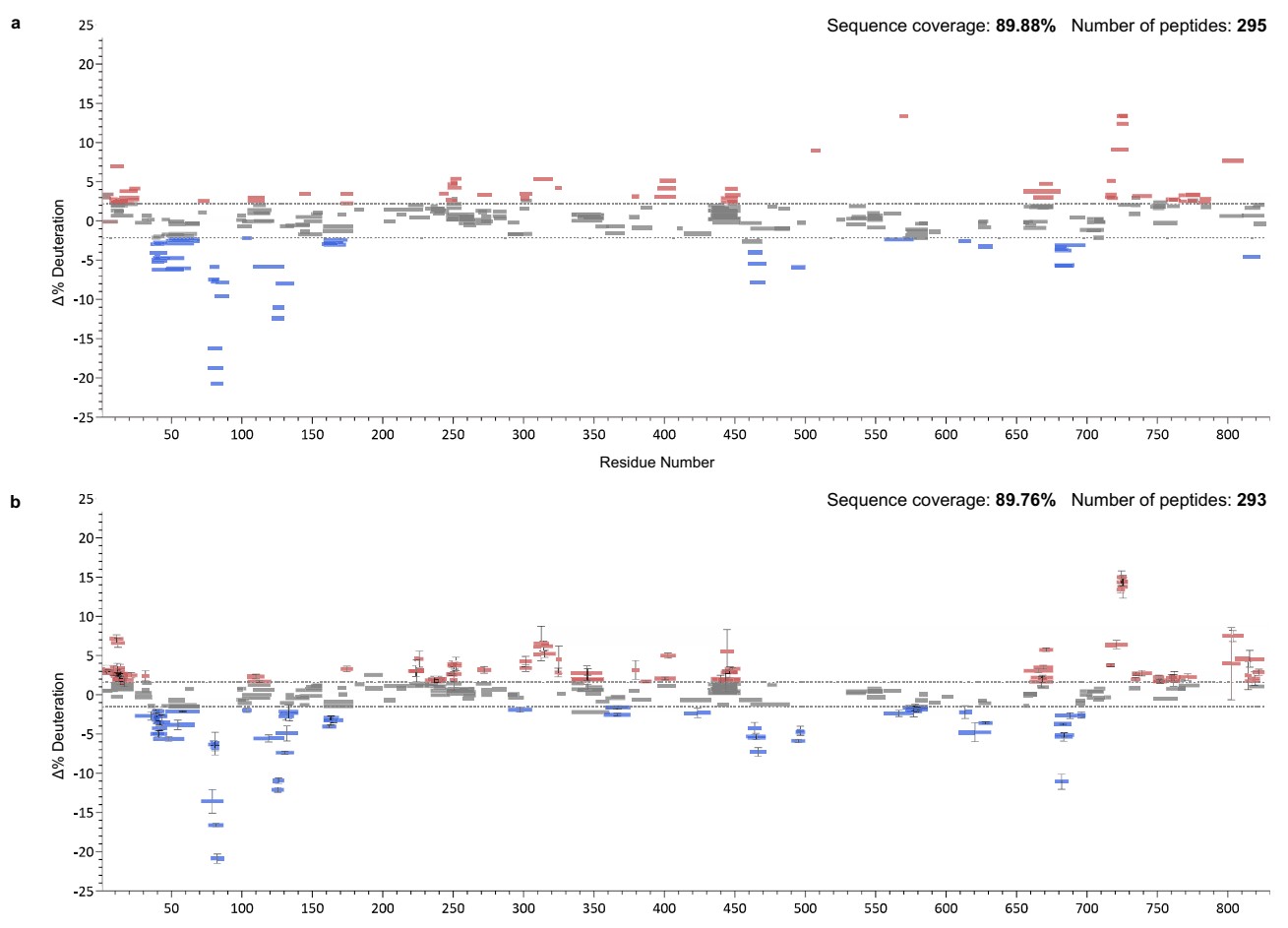

**Fig. 8 | Directly comparing manual and automated analysis. a** Woods plot produced through expert manual curation of MS[1] data. **b** Automated generation of Woods plot in AutoHX using both MS[1] and MS[2] data. Data compares novobiocin-bound Pol Θ to free Pol Θ. Residue numbering is based on pdb:5AGA. Red represents induced destabilizations and blue induced stabilizations. Data generated from *n* = 3 biologically independent samples for each of the two states and the error bars represent ± two standard deviations.

80% ACN. The flowrate was 15 µl/min for separation and 70 µl/min for loading and desalting. Data were collected using a standard HESI source. The spray voltage was set to 3500 V, sheath gas flow rate to 20, auxiliary gas flow rate to 7, and sweep gas flow rate to 0. The capillary temperature was set to 250 °C, the S-lens RF level to 65 and the auxiliary gas heater temperature to 80 °C. Approximate chromatographic peak widths were 13 s (FWHM) with this configuration.

DDA mapping runs were performed according to previously published optimized settings[46] with the top 12 ions selected for fragmentation. MS resolution was set to 70,000 with an AGC target of 5e5 and a 120 msec maximum trap fill time. The mass range was set to m/z 300–1040. MS[2] scans were collected at a resolution of 35,000 with an AGC target of 5e5 and a 120 msec maximum trap fill time. The isolation window was set to m/z 2.5 with a collision energy of 28 NCE using HCD. Total cycle time was ~1.8 s. Dynamic exclusion was set to 9 s to allow for approximately two MS[2] scans of each chromatographic feature.

DIA acquisitions of deuterated samples consisted of a single MS[1] scan of m/z 300–1040 followed by a set of 16 fragmentation bins with a width of m/z 50 and an overlap of m/z 4 per bin edge, covering the whole mass range. Full MS scan resolution was set to 70,000 with an AGC target of 1e6 and a 50 msec maximum trap fill time. DIA scans were set to a resolution of 35,000 with an AGC target of 2e5 and a 120 msec maximum trap fill time. The fixed first mass was set to m/z 200 and the collision energy to 28 NCE using HCD. Total cycle time was ~2.5 s.

## HX-MS[2] of Polθ ± novobiocin

**D₂O labeling.** For differential HX-MS[2] analysis, the ATPase domain of Polθ was produced in baculovirus-infected insect cells as previously described[47]. A 4 µM solution in HEPES buffer (25 mM, pH 7.4, 250 mM NaCl) was mixed 1:1 with 4 mM novobiocin (prepared in the same HEPES buffer with 2% DMSO) and incubated for 30 min. For each sample, 5 µL of the pre-incubated mixture was combined with 5 µL of D₂O-based HEPES buffer (25 mM, pD 7.4, 250 mM NaCl) to initiate deuterium labeling at room temperature. After 2 min of labeling, the reactions were quenched 1:1 (v/v) with quench/digestion buffer (500 mM glycine pH 2.3, 6 M urea) containing 0.6 µg/µL nepenthesin II digestion enzyme. Samples were digested at 8 °C for 2 min and then flash frozen in liquid nitrogen. Control samples without novobiocin were prepared with matched DMSO concentrations and processed as above, but with a quench/digestion buffer containing just 0.2 µg/µL nepenthesin II. All samples and controls were prepared in triplicate.

**Data collection.** Data were acquired on a Sciex TTOF 6600 instrument (with the Analyst TF 1.8.1 operating system) using an Optiflow Nano ESI Source, integrated with a Sciex Ekspert nanoLC 425 and a Trajan PAL HDX autosampler. Samples were manually injected into the cold compartment of the autosampler (set to 4 °C), outfitted with an Acclaim™ PepMap™ 100 C18 HPLC trap column for desalting (0.1 mm diameter, 5 µm particle size, 100 Å pore size, 20 mm length) at 10 µL/min mobile phase A for 3 min. The concentrated sample was

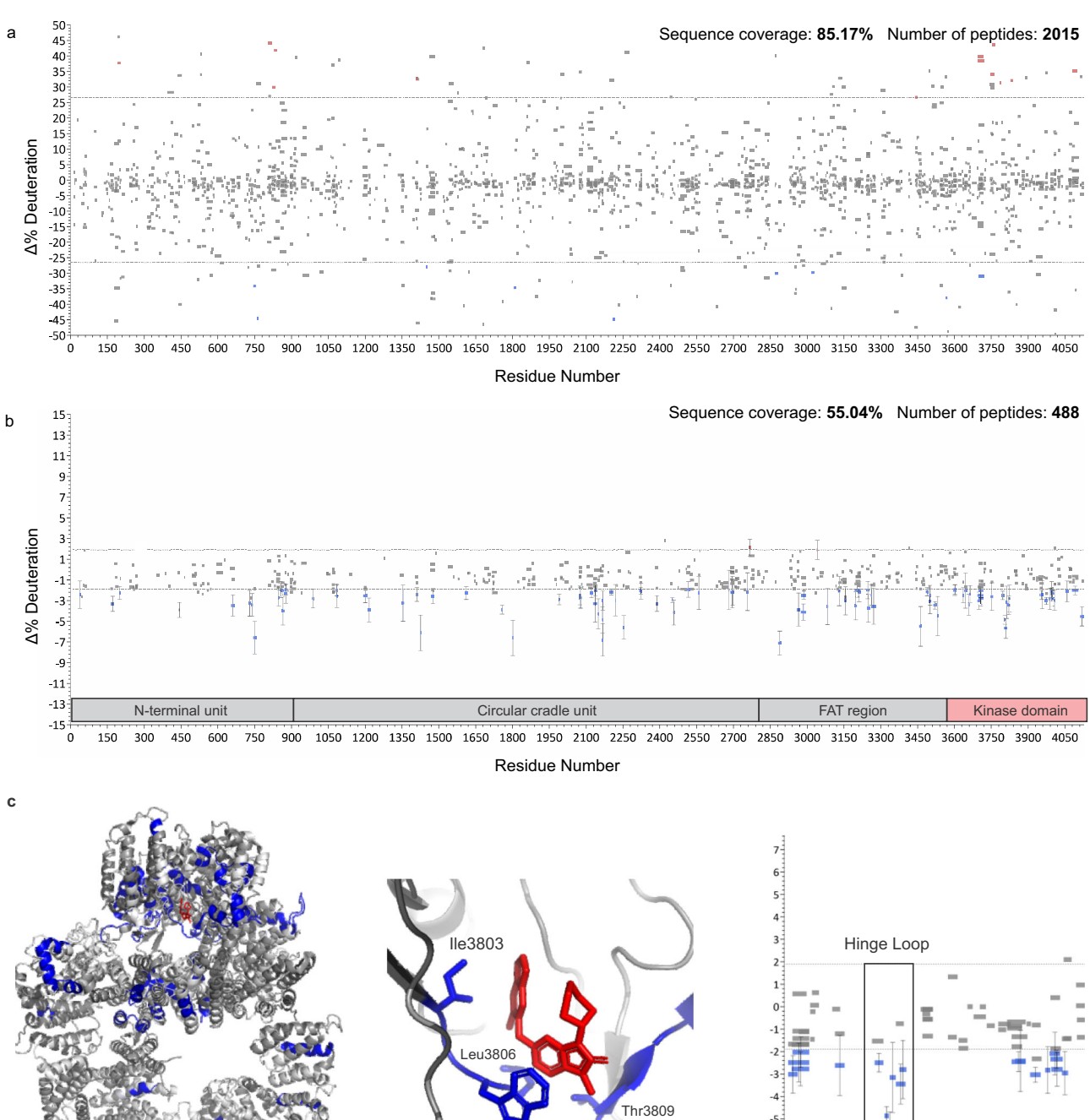

**Fig. 9 | Accelerated HX-MS analysis of DNA-PKcs isolated from low quantities of CHO cells. a** Unfiltered mapping file was blindly used for the generation of a Woods plot from replicate AZD7648-bound DNA-PKcs compared to a ligand-free control state. **b** Corresponding Woods plot produced using AutoHX without manual curation. **c** Mapping of high confidence stabilizations to PBB 7OTW, with expansion showing the highlighted hinge loop that defines the binding site. Key residues labeled as per Liang et al.[42], where blue residues represent stabilization, dark gray no change and light gray no coverage. Data generated from $n = 3$ biologically independent samples for each of the two states and the error bars represent ± two standard deviations.

eluted and separated on a nanoEase M/Z Peptide CSH C18 Column (75 μm diameter, 1.7 μm particle size, 130 Å pore size, 150 mm length), connected directly to the ion source, using a linear 10-min gradient from 5 to 35% mobile phase B at 250 nl/min. Source settings were as follows: GS1 = 7, GS2 = 0, CUR = 25, TEM = 0, ISVF = 3800. Approximate chromatographic peak widths were 6 s (FWHM).

DDA mapping runs were performed with the top 10 ions selected for fragmentation in "high sensitivity" mode. The MS[1] mass range was set to 400-850 m/z with a 150 ms accumulation time. The MS[2] scan range was set to 350–1100 m/z with an accumulation time of 120 ms and scans were collected using CID fragmentation with dynamic accumulation, dynamic collision energy setting, and dynamic

background subtraction functions enabled. The total cycle time was approximately 1.4 s. Dynamic exclusion was set to 20 s, allowing only a single MS$^2$ acquisition of all chromatographic features.

HX-MS$^2$ runs were performed with 16 variable DIA windows with m/z 3 overlap in "high sensitivity" mode. The MS$^1$ mass range was set to 400–850 m/z with a 200 ms accumulation time. DIA scan range was set to m/z 350–1100 with an accumulation time of 100 ms and scans were collected using CID fragmentation with a rolling collision energy, optimizing collision energy for each DIA window. The total cycle time was approximately 1.85 s.

### HX-MS$^2$ of DNA-PKcs ± AZD7648

**Coupling of anti-GFP nanobodies to magnetic beads**. 0.5 mg of Dynabeads™ (MyOne™ Streptavidin T1) were incubated with 10 μg of biotinylated Alpaca anti-GFP nanobody (ChromoTek, GTB-250) in a 100 μL incubation volume (PBS, pH 7.4, 0.1% Triton X-100) for 4 h at 4 °C. After conjugation, beads were washed three times with 100 μL of incubation buffer to wash away unbound nanobody.

**Expression and affinity enrichment of EGFP-DNA-PKcs construct.** Human EGFP-DNA-PKcs construct stably expressed in DNA-PKcs null V3 CHO cells was a kind gift from Dr. Kathy Meek (Michigan State University). The detailed preparation of V3 transfectant is described elsewhere[48]. The cells were cultured in 10 cm plates in α-MEM supplemented with 10% fetal bovine serum, 100 U/mL penicillin and streptomycin, 10 μg/μL cipromycin, and 10 μg/μL blasticidin. Cells were harvested by trypsinization and washed twice with 10 mL of PBS. Cell pellets were frozen at −80 °C until protein extraction. Frozen cell pellets were suspended in 1 mL of lysis buffer (50 mM Tris-HCl, 150 mM NaCl, 1 mM EDTA, 0.5% NP-40) with cOmplete™ protease inhibitor cocktail (Roche), phosphatase inhibitor cocktail (Roche), and universal nuclease (Pierce). The lysate was incubated on a nutator for 30 min at 4 °C followed by sonication with 3 × 5 s burst on ice. Lysate was centrifuged at 14,000×*g*, for 15 min at 4 °C, and protein concentration in clarified cell lysate was determined using BCA protein assay (Pierce). The protein concentration was adjusted to 2 mg/mL and 2 mg aliquots were flash frozen at −80 °C until pulldown. 80 μg of anti-GFP nanobody conjugated Dynabeads™ (~45 nL) were incubated with the cell lysate (4 mg of total protein content) prepared from the EGFP-DNA-PKcs expressing V3 CHO cells for 90 min at 4 °C. The beads were isolated and washed with 250 μL of PBS (pH 7.4) on a magnetic nano-isolator device described elsewhere. Beads were removed from the nano-isolator and collected in 8 μL HEPES buffer (10 mM, pH 7.4).

**D$_2$O labeling.** Prior to deuterium labeling, 10 μg of isolated beads (approximately 6 nL) were mixed in 4.5 μL of equilibration buffer (10 mM HEPES, pH 7.4) ± AZD7648 (Selleckchem cat # S8843, 1 μM) for 10 min. The deuterium labeling was carried out for 5 min by addition of 4.5 μL of D$_2$O labeling buffer (10 mM HEPES, pD 7.4). Labeling was quenched with addition of 1 μL of digestion buffer (500 mM Glycine-HCl, pH 2.3) containing 0.6 μg/μL of Nepenthesin II digestion enzyme. Digestion was carried out for 90 s at 10 °C. Following digestion, the beads were quickly magnetized (<5 s), and the digest was collected and flash frozen. All samples and controls were prepared in triplicate.

To determine the protein composition in the pulldown, 10 μg of isolated beads were incubated with MS-grade trypsin (15 ng, 50 mM AMBIC, pH 8.0) for an overnight on-bead digestion at 37 °C. The next morning, digest was quenched with the addition of 1 μL of 20% FA. The beads were magnetized (~20 s), and the digest was collected in a sampling vial for MS analysis.

**Data collection.** Data were collected on a prototype nanoHX ion source[16] coupled with an Orbitrap Eclipse (with the Xcalibur 4.6 operating system), outfitted with a Vanquish Neo loading pump and a Vanquish Neo gradient pump. Samples were manually injected into the chilled nanoHX source (held at 4 °C), which contained a PepMap™ Neo C18, 5 μm 300 μm x 5 mm trap cartridge (Thermo Fisher Scientific, P.N: 174500) and a PepMap™ Neo 2 μm C18, 75 μm × 150 mm analytical separation column (Thermo Fisher Scientific, P.N: DNV75150PN). Peptides were loaded and washed at 50 μl/min (0.4% formic acid) for 1 min. The concentrated sample was eluted using a linear 25-min gradient from 0%-40% mobile phase B at 300 nl/min. The spray voltage was nominally set to 1700V with the time-dependent feature enabled. The RF lens was set to 30 and the ion transfer tube to 270 °C. Approximate chromatographic peak widths were 6 s (FWHM).

DDA mapping runs were performed in OT/OT mode with the MS resolution set to 60,000 and a mass range of m/z 375–1000. MS$^2$ scans were collected at a resolution of 15,000 with isolation window set to m/z 1.6 with a collision energy of 30 NCE. Dynamic exclusion was set to 30 s after 2 occurrences within 15 s, to allow for approximately two MS$^2$ scans of each chromatographic feature. The AGC target was set to standard and maximum injection time to auto. The total cycle time was 1.5 s.

DIA acquisitions of deuterated samples consisted of a single MS$^1$ scan of m/z 350–1000 with a resolution of 60,000 followed by a set of 26 fragmentation bins with a width of m/z 25 and an overlap of m/z 4, covering the whole mass range with a resolution of 30,000. The fixed first mass in DIA was set to m/z 250 and the collision energy to 30 NCE. The AGC target was set to custom with a maximum injection time of 54 msec. The total cycle time was 2 s.

### Software design and availability

AutoHX functionality was built within the Mass Spec Studio 2.0 framework for integrative structural biology (version 2.4.0.3577)[49]. The software was written in C#, leveraging an extensive plugin-style repository of reusable content for rapid development of mass spectrometry applications. MSTools was used for workflow management[50].

### Reporting summary

Further information on research design is available in the Nature Portfolio Reporting Summary linked to this article.

## Data availability

All data are available via the PRIDE partner repository[51] with the dataset identifier PXD045012 (phosphorylase B, DNA-PKcs and Polθ). All processed data are available upon request from the corresponding author (D.C.S.).

## Code availability

A free compiled version of AutoHX with limited throughput capabilities is available at www.msstudio.ca, containing all the features applied in the described workflows.

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

## Acknowledgements
The authors would like to thank Dr. Kathy Meek, Michigan State University, for V3 cells stably expressing EGFP-DNA-PKcs. This work was funded by the Natural Sciences and Engineering Research Council of Canada Discovery Grants RGPIN 2017-04879 to D.C.S. and RGPIN-2019-04829 to S.P.L. J.A.T., and S.L.M. were funded in part by by National Institutes of Health (NIH) grant P01 CA092584, and J.A.T and A.S. were funded in part by R35 CA220430 and a Robert A.Welch Chemistry Chair (G-0010). Figures 1, 3, 5, and 7 were designed in BioRender.com under academic license.

## Author contributions
D.C.S., F.F., and V.S. conceptualized the project and designed experiments. F.F. collected and analyzed all mass spectral data. V.S. designed and wrote the software. D.A.C. assisted in coding and testing of software. M.F.K., P.D. and A.S. prepared and purified all proteins used in the study and in configuring the HDX experiments. D.C.S. and F.F. generated the first draft of the manuscript. S.C., R.V., J.A.T., and S.P.L. helped to finalize the paper.

## Competing interests
The authors declare no competing interests.
