## [Peer Review File · Nature Communications]

REVIEWER COMMENTS

Reviewer #1 (Remarks to the Author):

The manuscript by Filandr et al describes the development and application of HDX-MS methodology that utilizes MS/MS acquisition and automated data analysis of the deuteration of fragment ions to improve the confidence and throughput of the HDX-MS method. The approach is applied to measure the HDX of both a medium-sized purified protein, a large protein-ligand complex and the very large protein kinase DNA-PKcs extracted directly from a complex biological matrix (mammalian cells). Overall, I find the manuscript very exciting. The findings of the work are very interesting both in the context of improving throughput of the HDX-MS method when routinely applied to smaller-medium sized proteins and at the same time opening up for the method to better tackle non-purified protein samples and very complex protein systems. The workflow and software methodology should thus have widespread impact on the field. The manuscript is well written, experiments appear to be well designed and performed and the data is analyzed in considerable detail. Overall, I am highly positive wrt. publication and only have a few comments/suggestions to the authors:

Comments:

1. Line 70: "This restriction is no longer due to limitations in the analytical systems". While significant advancements have certainly been made, there are still numerous challenges with routine application of HDX-MS to such large systems. So I think this is somewhat over-stated and suggest the authors soften it up a bit.
2. Line 94: "The approach relies upon deuterium scrambling, which is ubiquitous under normal ion transmission conditions". A reference to a review on the topic should perhaps be included here as specific references only come later in the results section.
3. Line 260: "Scrambling is essentially complete using regular ion transmission settings as considerable ion activation occurs during desolvation and ion focusing". I think the authors should more clearly differentiate between scrambling occurring due to the type of MS/MS technique used – and scrambling occurring during ionization and transmission. For instance, it is well established that CID occurs with 100% scrambling – but I am not convinced that scrambling is "essentially complete" under "regular ion transmission settings" used in routine HDX-MS applications. To my mind, it is the fact that the authors use CID for MS/MS that allows them to convincingly assume that scrambling is 100% and thus enabling them to extract information concerning the deuteration of the precursor ion based on analysis of the fragment ions.
4. Line 261: "Only when transmission conditions are detuned and used in conjunction with electron-mediated fragmentation modes can scrambling be reduced". I think several studies have shown convincingly that scrambling can be eliminated under such conditions – not just reduced. See work from the Konerman, Griffin, Rand, Jørgensen labs.

5. Reference 36 and 37 are the same – and in general I think the referencing concerning the occurrence of scrambling during CID and its absence with ECD/ETD should be revised. There were several earlier works showing the same that should be referenced instead/also, including from the Jørgensen, Konerman and Rand labs.

6. As the authors have generated several large datasets on 3 proteins with HDX-CID-MS/MS data, they are in a unique position to confirm the absence of scrambling during CID across all sampled sequence variations. The authors should include a discussion of such an analysis, even if non-exhaustive. This will also serve to further validate the key assumptions made by their approach.

7. The authors should discuss how their data analysis workflow will handle EX1 type exchange kinetics and if any modifications of the workflow is needed, for instance, larger isolation widths etc. Also, the authors are advised to include a discussion of how their workflow should be adapted if a more conventional deuteration scheme was performed (i.e. labeling in 75-90% D₂O).

Reviewer #2 (Remarks to the Author):

Automating Data Analysis

Filandr et al.

The protein hydrogen exchange method analyzed by mass spec (HX-MS) has become a major biophysical methodology. Many approaches to analyzing the fairly complex HX-MS data, perhaps a dozen or so, have been put forward. In the continuing effort to adapt HX-MS to the study of ever larger proteins, Filandr et al present a new feature, upgraded from their previous version [Ref 28]. It incorporates HX-MS2 collision induced dissociation (CID) results, providing deuteration data from both MS1 and MS2 domains simultaneously into the data analysis, to more efficiently confirm peptide IDs and authenticate MS1-based deuteration calculations. The method presented may well be a useful addition to already available approaches.

Some additional information on certain points could be useful.

The authors state the two main benefits of doing HX-MS2 are:

1. Peptide validation: Since it still requires a peptide library from DDA runs, I wonder how much additional value of DIA data serves this purpose.
2. Data (deuteration measurement) authentication: A key question here is - what determines the width/shape of the normal distribution? i.e., we may need more focus and understanding of the linear scrambling model. For example, the authors state "fragments that undergo neutral loss appear to deviate from the linear model...". Do we know about the frequency of occurrence of neutral loss peaks? When a non-normal and/or wide distribution is observed, do we understand the possible causes (from experimental or data processing) rather than simply discarding from the dataset?

3. This normal distribution concept is also applied by the authors to the differential HDX analysis - testing the per-peptide significance. While it sounds reasonable and comparable to other existing significance tests (e.g. whole peptide-level deuteration; replicates), I do wish to read more reasoning and validation of this new test in statistics. Also, the authors write "The width of the distribution is used to assign a confidence interval to the change." How is it actually implemented and what's the recommended parameter setting? The statistics interpretation of the details should also be provided.

4. In Line #435, "... it revealed some hidden bias in manual review (e.g., assigning favorable changes in sequence regions already showing change)". I am unclear what this means. Some explanation and more details would be useful.

5. In Line #99, "We adapt computer vision algorithms ... ". Some more detail/explanation would be helpful.

6. As has been common in papers like this, there is no compare/contrast with any of the previously published data analysis methods. I suppose it may be asking too much given the labor that would be involved for any detailed comparison. Still I wonder if some general commentary might be included.

About Supporting Information

1. Fig. S1. RANSAC-based isotopologue filtering (as shown in Fig. S1) is a really good way the authors use to deselect spurious peaks.

2. The link in Line #155-158 is not working.

3. Fig. S2. The text font used throughout is illegibly small.

4. Labeling each peptide fragment by its amino acid sequence is unuseful. Their sequence positions, e.g. 1-20, would provide the information necessary to place and compare any one peptide with its overlapping neighbors.

5. The mass of each peptide is written to five significant figures. This is a childish error.

6. Although the % deuteration used as the Y axis is fairly common, it conceals the actual number of sites deuterated. Giving the actual number of deuterated sites would be better.

7. It is unfortunate that the time scale measured was so limited that in every case only a small fraction of each peptide was actually measured. This seems counter to the claim of much more efficient data acquisition.

SWEnglander

ZY Kan

Reviewer #3 (Remarks to the Author):

Reviewer #4 (Remarks to the Author):

General comments

Data analysis has been a major impediment in the HX-MS field, and with current instrument capabilities and digestion columns that generate hundreds of peptides from even simple proteins or protein complexes, manual curation of the data can take from weeks to months. This generally precludes the use of HX-MS for high-throughput applications, such as screening. In this study, the authors seek to tackle the manual validation burden by expanding their software (Mass Spec Studio) capabilities to leverage deuterium measurements from DIA acquisition-based approaches. MS2 level information serves in automatically validating the selection of peptides and deuterium measured content in minutes, improving confidence of generated results. This software capability is a nice continuation of previous papers by the same group, in which they introduced a DIA approach for confident deuterium measurements. The authors showcase the applicability of the software using data generated in different instrument platforms. The authors' work is notable in that they present a fully automated pipeline for analyzing HX-MS data, without the need for manual intervention. Overall, this is a very exciting and long-awaited advancement in the HX-MS field, especially since I have frequently found myself in the manual validation condition depicted in Figure 1. I highly recommend this for publication in Nature Communications. Minor revisions that will hopefully help in improving the manuscript are included below.

- Authors should include summary tables of all HDX data reported in this work, in both MS1 and DIA, as recommended in <https://pubmed.ncbi.nlm.nih.gov/31249422/>
- Figure 8: Panel a (unfiltered peptide map) of Figure 8 should be moved to the supplement, as it is now well-established that only a small fraction of peptides survive QC. The authors should keep panel a. in Figure 9, as it nicely illustrates the complexity of HX-MS data acquired and the amount of effort that would be required to validate such a dataset manually.
- What is missing from the paper are a few sentences describing the authors' experience with DIA for HX-MS measurements, which could serve as guidelines for new users. They comment briefly on a modest bias against peptides with relatively poor fragmentation (e.g., +1 peptides). I wonder if they can

comment on how the software handles lengthier peptides with b- or y- fragment ions of a higher charge state. Have different collision energies been evaluated on the same set of peptides, and what has been the result in terms of peptide survival? To better deconvolute busy m/z regions, do they see any benefit in employing DIA windows of variable width as opposed to fixed width? Can they remark on MS2 profile acquisition versus centroid acquisition?

Minor comments

Experimental Setup

Page 6, row 116: Authors should add the Phosphorylase B volume used per sample and final injection amount; was the same amount injected for DDA mapping and for DIA acquisitions? If not, this should be added for each acquisition

Page 7, row 150: DIA cycle time should be added for the QE+, as has been added in the DDA section

Page 10, row 220: Would it be helpful to provide the time it takes to magnetize the beads?

Page 11, row 247: Total cycle time should be added for DIA

Page 16, rows 335-338: Given that you can't use fragment ions from DIA experiments for localizing deuterium due to scrambling, I wonder whether the way redundancy is used here, causes confusion. The authors should clarify that these increase in redundancy does not translate into better resolution when it comes to D-measurements

Page 16, row 340: How is the 41% calculated? A more detailed explanation should be given

Below, I append a few suggestions for the authors on how to improve their software, but these are not required for the paper to be published.

- I tested the software in MS2 with both profile and centroid data, and it appears to function well. Running the Review version of the software successfully, however, required multiple trial-and-error attempts. To make it work, I had to eliminate the 0s sample for which I had a single replicate and maintain the same number of replicates across all other time points. I'm not sure whether this was a bug particular to the Review version, however, it didn't make any sense, as in real HX-MS experiments, the number of replicates may vary across time points.

- Further, it appears that the same non-deuterated RAW file cannot be used for different protein states. A new feature that would allow sharing the non-deuterated or FD controls across different states should be considered.

- It appears that b-, y-, and a- ions are currently being identified. The authors should consider incorporating internal ions, as these are very abundant, especially in Proline-containing peptides that have poor fragmentation

Reviewer comments are in blue and our responses are in black:

Reviewer #1

1. Line 70: “This restriction is no longer due to limitations in the analytical systems”. While significant advancements have certainly been made, there are still numerous challenges with routine application of HDX-MS to such large systems. So I think this is somewhat over-stated and suggest the authors soften it up a bit.

Granted. We wanted to acknowledge the collective progress made by the community, but it's true that more work is needed. We changed the sentence to “This restriction is not primarily due to limitations in the analytical systems.”

2. Line 94: “The approach relies upon deuterium scrambling, which is ubiquitous under normal ion transmission conditions”. A reference to a review on the topic should perhaps be included here as specific references only come later in the results section.

Done. We added an excellent review by Rand, Zehl and Jorgensen (*Acc. Chem. Res.* **2014**).

3. Line 260: “Scrambling is essentially complete using regular ion transmission settings as considerable ion activation occurs during desolvation and ion focusing”. I think the authors should more clearly differentiate between scrambling occurring due to the type of MS/MS technique used – and scrambling occurring during ionization and transmission. For instance, it is well established that CID occurs with 100% scrambling – but I am not convinced that scrambling is “essentially complete” under “regular ion transmission settings” used in routine HDX-MS applications. To my mind, it is the fact that the authors use CID for MS/MS that allows them to convincingly assume that scrambling is 100% and thus enabling them to extract information concerning the deuteration of the precursor ion based on analysis of the fragment ions.

We don't quite agree, but we are willing to alter the language a bit. A couple points first:

- In a paper by Rand *et al.* (**2011**, *JASMS*, 22,1784-93), they show in Figure 3 that ETD is nonpanacea. If source conditions are set at what I would consider typical settings for standard

proteomics to maximize sensitivity (what they call “harsh” settings, but only in a relative sense), full scrambling still happens even when they use ETD. I admit this is only one example peptide. However, it is widely used in the field to study these issues.

- We have tried very hard to minimize scrambling when using FAIMS devices (Thermo) and DMS devices (Sciex) and we’ve never been able to prevent it from happening (even when using ETD for Thermo and EAD for Sciex). This indicates that ion transmission has a big role to play in scrambling.

However, we think the following is a more balanced paragraph, and have updated the text accordingly:

“Scrambling is extensive when using ion optics settings consistent with high transmission efficiency. Such settings impart considerable ion activation during desolvation and ion focusing³⁷. Conventional CID fragmentation contributes additional thermal energy and ensures that scrambling is complete. Scrambling currently can only be overcome with a combination of milder ion transmission conditions together with electron-mediated fragmentation modes^{38,39}.”

4. Line 261: “Only when transmission conditions are detuned and used in conjunction with electron-mediated fragmentation modes can scrambling be reduced”. I think several studies have shown convincingly that scrambling can be eliminated under such conditions – not just reduced. See work from the Konerman, Griffin, Rand, Jørgensen labs.

Agreed. The text above makes the necessary change and we’ve added what we think are useful references to make the point. The inclusion of the 2014 *Acc. Chem. Res.* review on the topic in particular covers the many investigations into this topic.

5. Reference 36 and 37 are the same – and in general I think the referencing concerning the occurrence of scrambling during CID and its absence with ECD/ETD should be revised. There were several earlier works showing the same that should be referenced instead/also, including from the Jørgensen, Konerman and Rand labs.

We added three more references to the paragraph above, which looks pretty comprehensive now.

6. As the authors have generated several large datasets on 3 proteins with HDX-CID-MS/MS data, they are in a unique position to confirm the absence of scrambling during CID across all sampled sequence variations. The authors should include a discussion of such an analysis, even if non-exhaustive. This will also serve to further validate the key assumptions made by their approach.

It is good commentary for the conclusion, so we have added a bit of text there. But note that we really can’t comment very extensively. We don’t know the deuterium localization of these peptides before scrambling. We can assume that unscrambled peptides would deviate (to varying degrees) from the linear fit that our software looks for. In extreme cases like the “Jørgensen” peptide, we might see a divergence of b and y ions (akin to two linear fits). We went through all the peptides manually and didn’t see any evidence of this. Both b and y ions seem to normally scatter around the fitted deuteration trend line. We revised the text in the beginning of the conclusion as follows:

“Automation tools have improved the rate at which HX-MS data can be collected, but the burden of manual data analysis has limited the extent to which the technology can be applied to many interesting problems. Deuterium scrambling offers a solution to the challenge of automating data analysis, by turning a problem into an opportunity. A manual inspection of the fragment models for all the peptides in the study did not show any evidence for retaining deuterium localization. Scrambling indeed seems to be the norm. By invoking DIA methodology for comprehensive

fragment detection, AutoHX removes the burden of data analysis while simultaneously providing both peptide validation and data authentication.”

7. The authors should discuss how their data analysis workflow will handle EX1 type exchange kinetics and if any modifications of the workflow is needed, for instance, larger isolation widths etc. Also, the authors are advised to include a discussion of how their workflow should be adapted if a more conventional deuteration scheme was performed (i.e. labeling in 75-90% D₂O).

Yes, good points to make, and it provides us with another opportunity to encourage the community to move away from labeling with high levels of D₂O! We added the following comments:

“The degree of overlap used in the DIA experiment is selected based on the percentage of D₂O used in the experiment. A window overlap of m/z 4 was found to be sufficient for 50% D₂O labeling and did not significantly compromise cycle times on either the TOF or Orbitrap platforms used in this study. We note that more common labeling levels of 75-90% D₂O would require slightly larger overlaps (e.g. m/z 5-6) but we note parenthetically that such high percentages are not a requirement of the basic HX-MS technique⁴².” (Page 14, lines 303-308)

Regarding EX1, we mentioned on page 16 that AutoHX can also be accommodated, and have added a statement that the chosen overlap would also be impacted:

“EX1 kinetics would also be carried over to the fragments in MS² space, and in extreme cases (e.g., a large peptide with high deuteration and labeling at 90% D₂O) the DIA window overlap may need to be widened slightly.” (Page 16, lines 335-337)

Reviewer #2

1. The authors state the two main benefits of doing HX-MS2 are:
 - a. Peptide validation: Since it still requires a peptide library from DDA runs, I wonder how much additional value of DIA data serves this purpose.

First, to be clear on how we use the terms, so we are discussing the same things: “Validation” is the confirmation of a peptide’s identity and “authentication” is the confirmation of the peptide’s deuteration value. The text was adjusted to make the use of these terms clearer. We now describe the conventional process of filtering peptides and checking and adjusting peptide deuteration calculations as “curation”, to avoid any confusion of terms.

The benefit of DIA for validation becomes more obvious with the more complex sample types. The library is necessary to create the library of all peptides potentially found in the deuterated data. The authentication comes in when searching the deuterated data. Currently HX-MS software only uses the peptide mass and retention time to search for any given peptide in the deuterated data. Instances of misassignment are known to occur and we have indeed found evidence of this in our data. The presence of fragments is a reliable way to eliminate instances of this misassignment, as the wrongly assigned peptide will not produce expected fragments and will be filtered out in our data processing step.

2. Data (deuteration measurement) authentication: A key question here is - what determines the width/shape of the normal distribution? i.e., we may need more focus and understanding of the linear scrambling model. For example, the authors state "fragments that undergo neutral loss appear to deviate from the linear model...". Do we know about the frequency of occurrence of neutral loss peaks?

When a non-normal and/or wide distribution is observed, do we understand the possible causes (from experimental or data processing) rather than simply discarding from the dataset?

The width and shape of the distribution will be determined by a combination of ion statistics (i.e., peak intensity), the size of the peptide (number of exchangers), and the degree of overlap in the MS¹ and MS² space. Basically, all the things that would impact a useful deuteration measurement. The tighter the normal distribution, the more reliable the measurement. As we collect more and more data, we can begin to place tighter expectations on the nature of the distribution, but at some level the quality will always be statistically assessed. Finally, one reason for the non-linear behaviour of the neutral loss fragments may be a kinetic isotope effect as we suggested in an earlier publication (Percy AJ et al, **2009**, *Anal. Chem*, 81, 7900-7907).

3. This normal distribution concept is also applied by the authors to the differential HDX analysis - testing the per-peptide significance. While it sounds reasonable and comparable to other existing significance tests (e.g. whole peptide-level deuteration; replicates), I do wish to read more reasoning and validation of this new test in statistics. Also, the authors write "The width of the distribution is used to assign a confidence interval to the change." How is it actually implemented and what's the recommended parameter setting? The statistics interpretation of the details should also be provided.

What we are doing here is fundamentally no different than what any HDX practitioner does in deuteration analysis: we take replicate measurements and determine an average and a standard deviation. In conventional MS¹-only analyses, the statistics we use simply assume a normal distribution, but we only have 2-4 replicates so our ability to populate a distribution of values is actually limited. But as figure 4 shows, the massive increase in redundancy allows us to sample the distribution of values much more effectively. It is no more complicated than that. Note that we state (in the Woods plots for differential analysis) that the error bars represent a confidence interval of 95% (approximately +/- 2 standard deviations of this distribution).

4. In Line #435, "... it revealed some hidden bias in manual review (e.g., assigning favorable changes in sequence regions already showing change)". I am unclear what this means. Some explanation and more details would be useful.

It is related to our comments to your point 1: mining only MS¹ space, we assumed that peptides at a given m/z and retention time were correctly identified if they were in our library. It turns out that some of the peptides were not supported with fragment data and were incorrectly assigned during manual review. We added some clarity to this section as follows:

"For example, we detected changes in sequence regions already showing change, but were obviously less diligent in their assessment. With the benefit of MS² data, we could see that some peptides were misidentified as they were not supported by fragment data." (Page 21, lines 447-449)

5. In Line #99, "We adapt computer vision algorithms ... ". Some more detail/explanation would be helpful. Done. We indicated that this is a revised RANSAC algorithm, but we kept it brief as this is the introduction. A few more details are already in the body of the text (top of page 15).

6. As has been common in papers like this, there is no compare/contrast with any of the previously published data analysis methods. I suppose it may be asking too much given the labor that would be involved for any detailed comparison. Still, I wonder if some general commentary might be included.

A direct comparison with previously published data analysis software is not possible, because no other software is able to analyze DIA MS² data for the purposes of deuteration assignment. The comparison of the typical data analysis workflow (manual validation) with our new automated approach is sufficiently demonstrated in Figure 8. We used the publicly available version of the HX-DEAL module in Mass Spec Studio 2.0 to analyze just the MS¹ traces of our data for this comparison.

7. About Supporting Information

- a. Fig. S1. RANSAC-based isotopologue filtering (as shown in Fig. S1) is a really good way the authors use to deselect spurious peaks.

Thank you!

- b. The link in Line #155-158 is not working.

They seem to work now.

- c. Fig. S2. The text font used throughout is illegibly small.

Figure S2 has been resized to make it more legible.

- d. Labeling each peptide fragment by its amino acid sequence is unuseful. Their sequence positions, e.g. 1-20, would provide the information necessary to place and compare any one peptide with its overlapping neighbors.

That is a very good point. Figure S2 has been reworked to include amino acid sequences and sorting of the peptides was adjusted accordingly to allow for a better regional comparison.

- e. The mass of each peptide is written to five significant figures. This is a childish error.

Corrected.

- f. Although the % deuteration used as the Y axis is fairly common, it conceals the actual number of sites deuterated. Giving the actual number of deuterated sites would be better.

As you note, what we did is common practice. It faithfully reports the actual deuterium content, with no assumptions.

- g. It is unfortunate that the time scale measured was so limited that in every case only a small fraction of each peptide was actually measured. This seems counter to the claim of much more efficient data acquisition.

We are not sure what the reviewer means here. A large fraction of the peptides we show in Figure S2 approach 50% or more total deuteration, and more importantly we measure 6 timepoints in replicate. Although we didn't go to 24 hours (for example), we are making at least as many timepoint measurements as is done in the literature for such activities.

Reviewer #4

1. Authors should include summary tables of all HDX data reported in this work, in both MS1 and DIA, as recommended in <https://pubmed.ncbi.nlm.nih.gov/31249422/>

As one of the senior corresponding authors on that paper, I don't agree. It was not our intention to impose these criteria on papers that publish new methods. This was an important point for the group of

authors during the writing of that paper, to ensure that innovations are not constrained. At any rate, the text of our current manuscript does make all the salient points from the recommendations paper.

2. Figure 8: Panel a (unfiltered peptide map) of Figure 8 should be moved to the supplement, as it is now well-established that only a small fraction of peptides survive QC. The authors should keep panel a. in Figure 9, as it nicely illustrates the complexity of HX-MS data acquired and the amount of effort that would be required to validate such a dataset manually.

A good point. We simplified Figure 8 (two panels) and created a new Figure S3 to showcase the unfiltered map.

3. What is missing from the paper are a few sentences describing the authors' experience with DIA for HX-MS measurements, which could serve as guidelines for new users. They comment briefly on a modest bias against peptides with relatively poor fragmentation (e.g., +1 peptides). I wonder if they can comment on how the software handles lengthier peptides with b- or y- fragment ions of a higher charge state. Have different collision energies been evaluated on the same set of peptides, and what has been the result in terms of peptide survival? To better deconvolute busy m/z regions, do they see any benefit in employing DIA windows of variable width as opposed to fixed width? Can they remark on MS2 profile acquisition versus centroid acquisition?

Most of the optimizations will be dependent on the sample being analyzed. That is, is the sample simple or complex? If complex, then perhaps a more tailored approach to DIA windowing will be in order, just like for conventional DIA experiments for proteomics samples. We are currently exploring this tailoring on new projects. Collision energies are an interesting topic, as these are applied to the total window contents, regardless of charge state. In this respect, stepped collision energies would be a good idea to use. There is no problem with longer peptides that we can tell. They generate more fragments and make sampling of the deuterium distribution much more extensive. Regarding centroid vs. profile, this will always be platform dependent.

We added a couple of statements on page 23 at the end of the DNA-PK section to get readers thinking about such things:

Further optimization of the isolation should enhance this HX-MS assay and support the expansion of screening activities, and tailored DIA methods (e.g., variable window sizes and stepped collision energies) should enhance sequence coverage.

4. Minor comments

- a. Experimental Setup, Page 6, row 116: Authors should add the Phosphorylase B volume used per sample and final injection amount; was the same amount injected for DDA mapping and for DIA acquisitions? If not, this should be added for each acquisition.

Yes, the same amounts were injected for DDA and DIA. We added sample volumes to the experimental section.

- b. Page 7, row 150: DIA cycle time should be added for the QE+, as has been added in the DDA section.

Fixed. The QE+ DIA cycle times were added.

- c. Page 10, row 220: Would it be helpful to provide the time it takes to magnetize the beads?

The times to magnetize the beads were added to the experimental section.

- d. Page 11, row 247: Total cycle time should be added for DIA

Fixed. The Eclipse DIA cycle time was added.

- e. Page 16, rows 335-338: Given that you can't use fragment ions from DIA experiments for localizing deuterium due to scrambling, I wonder whether the way redundancy is used here, causes confusion. The authors should clarify that these increase in redundancy does not translate into better resolution when it comes to D-measurements.

We modified the text to make it clear the increased redundancy does not translate into increased resolution.

"This extra redundancy does not improve resolution, but it does translate into higher-quality measurements." (Page 16, line 346-347)

- f. Page 16, row 340: How is the 41% calculated? A more detailed explanation should be given

We changed the sentence for more clarity:

"For example, in the Phosphorylase B dataset, including the MS² data decreases the standard deviation of the measured peptide deuteration by 41% compared to using just MS¹ data." (Page 16, line 347-349)

- 5. Below, I append a few suggestions for the authors on how to improve their software, but these are not required for the paper to be published.
 - a. I tested the software in MS2 with both profile and centroid data, and it appears to function well. Running the Review version of the software successfully, however, required multiple trial-and-error attempts. To make it work, I had to eliminate the 0s sample for which I had a single replicate and maintain the same number of replicates across all other time points. I'm not sure whether this was a bug particular to the Review version, however, it didn't make any sense, as in real HX-MS experiments, the number of replicates may vary across time points.
 - b. Further, it appears that the same non-deuterated RAW file cannot be used for different protein states. A new feature that would allow sharing the non-deuterated or FD controls across different states should be considered.
 - c. It appears that b-, y-, and a- ions are currently being identified. The authors should consider incorporating internal ions, as these are very abundant, especially in Proline-containing peptides that have poor fragmentation

Thank you for the suggestions. We are currently rebuilding the front-end elements of the workflow as they are a bit dated and will keep these in mind. Internal ions are a good idea, as we do mine them in our crosslinking app (so the technology exists in the Studio).

REVIEWERS' COMMENTS

Reviewer #1 (Remarks to the Author):

The authors have nicely addressed my main comments.

I have just one suggestion for the new sentence in the beginning of the conclusion (to my comment no. 6):

"A manual inspection of the fragment models for all the peptides in the study did not show any evidence for retaining deuterium localization."

The phrase "retaining deuterium localization" is a bit unspecific to me - I think it would be more clear to write "retaining regioselective labeling" or "retaining a non-scrambled labeling pattern" or some such.

Reviewer #2 (Remarks to the Author):

The paper now seems to me to be acceptable for publication.

Reviewer #3 (Remarks to the Author):

Reviewer #4 (Remarks to the Author):

All of my points have been addressed. I highly recommend this article for publication.

Malvina Papanastasiou

Reviewer #4 (Remarks on code availability):

I was able to install and run the code with data generated in-house and have incorporated comments in my first review. The code works as the authors have described it in the paper.

Reviewer #1

1. The authors have nicely addressed my main comments. I have just one suggestion for the new sentence in the beginning of the conclusion (to my comment no. 6): "A manual inspection of the fragment models for all the peptides in the study did not show any evidence for retaining deuterium localization." The phrase "retaining deuterium localization" is a bit unspecific to me - I think it would be more clear to write "retaining regioselective labeling" or "retaining a non-scrambled labeling pattern" or some such.

We changed the wording to "retaining regioselective labeling" as suggested.